# Silicon Foliar Spray and Substrate Drench Effects on Plant Growth, Morphology, and Resistance to Wilting with Container-Grown Edible Species

**Joshua B. Tebow, Lauren L. Houston and Ryan W. Dickson ***

Department of Horticulture, University of Arkansas, Fayetteville, AR 72701, USA; jbtebow@uark.edu (J.B.T.); llhousto@uark.edu (L.L.H.)
* Correspondence: ryand@uark.edu; Tel.: +1-479-575-2533

**Abstract:** The objective of this study was to evaluate silicon (Si) foliar spray and substrate drench effects on plant growth and morphology for container-grown edible crops during greenhouse production, as well as resistance to plant wilting during post-production. In the first greenhouse experiment, basil received Si foliar sprays at 0, 50, 100, 200, and 400 mg·L$^{-1}$ Si. In the second greenhouse experiment, Si was applied as either a foliar spray (500 mg·L$^{-1}$ Si) or substrate drench (100 mg·L$^{-1}$ Si) with six edible crop species. Supplemental Si increased shoot Si levels but had minimal effects on plant growth and morphology, except for parsley, which resulted in distorted growth and phytotoxicity. In the first experiment, 200 and 400 mg·L$^{-1}$ Si foliar sprays increased plant resistance to wilt by 2.2 and 2.5 d, respectively; however, this was not observed in the second experiment. All species accumulated Si with the control (no Si) treatments, indicating trace amounts of Si were taken up from the substrate, fertilizer, spray surfactant, and irrigation water. Only cucumber was classified as a Si "accumulator" with a high capacity for Si uptake. Results emphasize the need to conduct preliminary trials with supplemental Si to avoid issues of phytotoxicity.

**Keywords:** basil; cucumber; parsley; rosemary; thyme; tomato; silicon; container crop; soilless substrate



## 1. Introduction

Silicon (Si) is a major constituent of many mineral field soils, and although not considered an essential element in plant nutrition, it has been reported to have beneficial effects on plant growth during crop production [1–3], particularly under conditions leading to plant stress. For some plant species, Si uptake increases resistance to infection by plant pathogens, damage from ultraviolet light and extreme temperatures, certain physiological and nutritional disorders, drought, and wilting [2–6]. Field and soil-grown crops typically accumulate Si [2], but plant tissues can vary considerably in Si concentration ranging from 1 to 100 mg Si·g$^{-1}$ dry weight depending on plant species and growing conditions. Plants grown in containers using soilless culture accumulate low amounts of Si compared to field production [3,7], primarily because the soilless growing substrates, applied fertilizers, and irrigation water sources used for container production are typically low in Si.

Past research with container-grown floriculture species has shown that supplemental Si applications during production can reduce disease pressure and increase plant quality [3,7–11], where Si is typically supplied by incorporation into the growing substrate, dissolving into the applied fertilizer solution, or as a foliar spray. Supplemental Si applications with container crops have been reported to decrease incidence of foliar fungal diseases such as powdery mildew, black spot disease, and botrytis [3–5,12,13], as well as root rot fungal diseases such as *Pythium* [14]. McAvoy and Bible [15] showed weekly Si foliar sprays increased tissue Si concentrations and the distribution of calcium in poinsettia bracts (*Euphorbia pulcherrima* Willd. Ex. Klotzsch), decreasing susceptibility to the common physiological and calcium-related disorder known as bract-edge burn. Frantz et al. [16]

and Kamenidou et al. [9] found supplemental Si applications increased leaf resistance to transpiration and delayed wilting (i.e., increased resistance to wilt) for container-grown poinsettia, sunflower (*Helianthus annuus* L.), and zinnia (*Zinnia elegans* Jacq.). These beneficial effects of Si are commonly observed in stress conditions which can all occur in large scale production systems.

Production of container-grown edible crops, such as vegetables and culinary herbs, for grocery and retail supermarkets is an increasing trend among greenhouse floriculture operations in the USA [17,18]. In contrast to floriculture crops, the use of pesticides is undesirable and more restricted for edible crops intended for human consumption, and growers have fewer options to control plant pathogens and mitigate disease losses during production. In addition, common practice for container-grown edibles is to deliver a final irrigation at the end of production, after which plants are packaged in plastic sleeves and shipped to retail [19]. Because plants are not typically irrigated once in the plastic sleeves, wilting is a major contributor to poor plant quality in the retail environment, leading to decreased consumer satisfaction and sales.

Supplemental Si applications during production may be a strategy for growers to mitigate the incidence of disease as well as delay wilting and extend the shelf-life of container edibles in retail. The majority of Si research for container crops focuses on Si foliar sprays or substrate drenches with floriculture species [7], and little published information exists regarding Si effects on container-grown vegetables and culinary herbs. Despite the potential benefits, plant species are also known to differ in Si uptake and tolerance to supplemental Si applications, which can sometimes impact plant quality and cause changes in plant morphology [7–9].

The objective was to evaluate the effects of Si foliar sprays and substrate drenches on Si uptake and plant growth and morphology during greenhouse production with container-grown edibles, as well as plant resistance to wilting in a simulated indoor retail environment. We hypothesized Si foliar sprays and drench applications would increase plant tissue Si concentrations but have minimal impact on plant growth and morphology. We also hypothesized that edible plant species would differ in Si uptake and accumulation in tissues.

## 2. Materials and Methods

### 2.1. Experiment #1: Evaluation of Si Foliar Spray Concentrations with Basil

A single-factor experiment evaluated the effects of Si foliar spray applications on plant growth and quality during production as well as resistance to wilting during post-production for container-grown basil (*Ocimum basilicum* 'Compact Genovese' L.). The experiment took place in a polycarbonate and controlled-environment greenhouse located at the University of Arkansas in Fayetteville, AR (36.0822° N, 94.1719° W). Average daily temperature and daily light integral during the experiment were (mean ± standard deviation) $22.7 \pm 1.1$ °C and $19.3 \pm 7.2$ mol·m$^{-2}$·d$^{-1}$, respectively.

Pelleted basil seed were sown into 128-cell plug tray at one pellet per cell and germinated in soilless peat-based substrate (ProMix BX; PremierTech, Deslon, QC, Canada) on a greenhouse bench. At the 2-true leaf stage, seedlings were thinned to one plant per plug and transplanted into 10 cm standard pots (Poppelmann Plastics, Claremont, NC, USA) containing ProMix BX at two plugs per pot. For the duration of the experiment, all treatment replicate plants were irrigated uniformly with fertilizer solution once substrate moisture dropped to approximately 50% of container capacity, using a commercial water-soluble fertilizer (17-3-17 Peters Peat-Lite Special®; Everiss, Geldermalsen, The Netherlands) mixed at 150 mg·L$^{-1}$ nitrogen (N) in tap water with an electrical conductivity of <0.3 mS·cm$^{-1}$ and <60 mg·L$^{-1}$ bicarbonate alkalinity. Each replicate plant for each treatment received 1.35 L of fertilizer solution and 202.5 mg of N.

The experiment started with the first Si foliar spray treatment, made 3 days after transplant. Foliar spray treatments were then made every 7 d, and the experiment lasted a total of 45 d. Treatments consisted of spraying solution containing 0, 50, 100, 200, and 400 mg·L$^{-1}$ Si mixed with reagent-grade sodium silicate in deionized water. Solutions

also contained a nonionic surfactant (Aquatrols, Paulsboro, NJ, USA) at 0.3 mL·L$^{-1}$ to reduce water tension and increase Si absorption across leaf surfaces. A 0 mg·L$^{-1}$ Si plus no surfactant spray (100% deionized water) was also applied as an additional control treatment. Each spray application occurred between 18:00 and 20:00 h, and plants were sprayed to runoff.

The experiment contained six treatment levels (five Si spray concentrations plus a 100% deionized water control) arranged using a randomized complete block design with four blocks. Each containerized basil plant was one replicate, with two replicates per treatment per block. At the end of greenhouse production at 45 d, four replicates per treatment were destructively sampled for plant growth and quality measurements. The remaining four replicates per treatment continued for post-production evaluation of resistance to wilting.

At the end of production, leaf SPAD chlorophyll content was measured using a Minolta leaf SPAD chlorophyll index meter, where each replicate measurement was the average of six measurements taken on randomly selected leaves per plant. Canopy height was measured from the substrate surface to the tallest shoot tip. Canopy width measurements consisted of taking the average of two canopy width measurements collected at 90° and perpendicular angles. Shoots were cut at the substrate surface, weighed for shoot fresh weight determination, and then placed in a drying oven for 3 d. After the 3 d, the plants were reweighed for shoot dry weight determination. Dry shoot tissue samples (200 mg per sample) were dry ashed in a muffle furnace at 500 °C for 5 h, solubilized in 0.5 *N* HCl, and analyzed for Si concentration using inductively coupled plasma emission spectrophotometry (University of Florida IFAS Analytical Services, Gainesville, FL, USA).

Resistance to wilting during post-production was measured as the number of days until visible plant wilt in a simulated indoor retail environment, determined for four replicates per treatment. At the end of production, replicate plants were irrigated to container-capacity, weighed, and moved to a post-production room with a 12-h photoperiod and temperature ranging from 20 to 25 °C. Plants were checked daily for wilting between 10:00 and 11:00 h. Date and total weight were recorded at visible wilt for each replicate, which was characterized by the loss of leaf and stem turgidity, dull coloration of foliage, and the downward pointing of the apical shoot tips. Total plant water loss was calculated by subtracting the total weight at visible wilting from the total weight at container capacity for each replicate. Daily evaporation was also measured using evaporation pans placed at the edges as well as randomly within the research plot. To minimize the variability in plant water loss caused by fluctuations in the retail environment temperature, the number of days until visible wilt was standardized by dividing the total water loss per replicate by the average daily water loss from evaporation pans.

The effects of Si foliar spray applications on basil growth during production and days until visible wilt during post-production were evaluated using analysis of variance (ANOVA) with PROC GLIMMIX in SAS 9.4 (SAS Institute, Cary, NC, USA). There were no differences between the 0 mg·L$^{-1}$ Si surfactant and no-surfactant control treatments, and therefore control replicates were combined for greater statistical power. The ANOVA was significant for several variables with clear positive trends in plant responses to increasing Si level; however, there lacked statistical differences between treatment means using common multiple comparison tests such as Tukey's honestly significant difference (hsd) at $\alpha = 0.05$. Single degree-of-freedom contrasts ($\alpha = 0.05$) were therefore used as a simple and robust post hoc statistical analysis comparing each Si treatment effect to that of the combined non-silicon controls.

### 2.2. Experiment #2: Edible Plant Species Supplied with Si Foliar Sprays and Substrate Drenches

A factorial experiment evaluated Si foliar spray and substrate drench treatment effects on plant growth and morphology and Si uptake for six common container-grown edible species. Edible plant species included basil (*Ocimum basilicum* 'Genovese' *L.*), cucumber (*Cucumis sativus* 'Straight Eight' *L.*), parsley (*Petroselinum crispum Mill.*), rosemary (*Salvia rosmarinus L.*), thyme (*Thymus vulgaris L.*), and tomato (*Solanum lycopersicum* 'Supersweet'

*L.*). The experiment was conducted in a polycarbonate controlled-environment greenhouse in Fayetteville, AR (36.0822° N, 94.1719° W). Average daily temperature during the experiment was (mean ± standard deviation) 22.8 ± 0.4 °C and daily light integral was 13.6 ± 8.0 mol·m$^{-2}$·d$^{-1}$ of photosynthetically active radiation.

Seedlings of basil, cucumber, parsley, thyme, and tomato and vegetatively propagated tip cuttings of rosemary were grown in 128-count trays and then transplanted into 11.5 cm diameter square containers (The HC Companies, Twinsburg, OH, USA) with a peat-based soilless substrate (ProMix BX; Premier Tech, Quebec, Canada) at one plant per container. Young plants were well rooted and of the pullable plug stage at transplant. Plants were positioned on a greenhouse bench on 30.7 cm center spacing, and plastic saucers were placed under each container to collect leachate and allow for reabsorption into the substrate.

At transplant and for each subsequent irrigation event, each container received 150 mL of water-soluble fertilizer (17-3-17 Peters Peat-Lite Special®; Everiss, Geldermalsen, The Netherlands) mixed at 150 mg·L$^{-1}$ N in municipal tap water. The tap water contained an electrical conductivity (EC) of 0.20 mS·cm$^{-1}$ and <60 mg·L$^{-1}$ bicarbonate alkalinity. Silicon concentrations in the tap water and applied fertilizer solution were 2.99 and 3.01 mg·L$^{-1}$ Si, respectively, and the substrate solution contained 6.88 mg·L$^{-1}$ Si using a saturated media extract method [20]. Each container received 1000 mL of applied fertilizer solution during the experiment, which provided 17 mg N and 6.88 mg Si.

The experiment started at transplant, and Si treatments started 7 days after transplant and were applied every 7 days thereafter between 18:00 and 20:00 h. The experiment was a 3 × 6 factorial with Si application (Si foliar spray, Si substrate drench, non-Si control) and plant species (basil, cucumber, parsley, rosemary, thyme, and tomato) as factors. Treatments were arranged using a randomized complete block design with three blocks and one treatment replicate per block, where each replicate was one containerized plant. The last Si treatments were applied 28 days after transplant, for a total of four treatment applications per replicate.

Silicon foliar treatments consisted of spraying each plant until runoff with potassium silicate solution (AgSil 21®, PQ Corporation, Valley Forge, PA, USA) mixed at 500 mg·L$^{-1}$ Si in deionized water and with a surfactant at 0.4 mL·L$^{-1}$ (Aquatrols, Paulsboro, NJ, USA). The 500 mg·L$^{-1}$ Si was the lower recommended rate for Si foliar applications according to the AgSil® product label. Spray volumes per replicate differed between species and increased over time with plant growth. Total Si foliar spray volumes (in mL) and mg of Si (in mg) applied per replicate were 38.0 and 19.0 for basil, 57.3 and 26.8 for cucumber, 38.0 and 19.0 for parsley, 18.0 and 9.0 for rosemary, 21.7 and 10.8 for thyme, and 56.3 and 28.2 for tomato, respectively.

Silicon drench treatments consisted of applying 150 mL of the same potassium silicate solution to the substrate mixed at 100 mg·L$^{-1}$ Si in deionized water and without surfactant. The 100 mg·L$^{-1}$ Si was the recommended rate for Si drenches according to the AgSil® (Certis USA, Columbia, MD, USA) product label. A total of 600 mL of Si drench solution and 60 mg of Si were applied per replicate container during the experiment.

For the non-Si control treatments, deionized water was applied every 7 days as both a foliar spray (with surfactant) and substrate drench. In addition, plants which received the Si foliar and drench treatments also received a deionized water drench or foliar spray application, respectively, to minimize any potential biases in plant growth caused by saturating the root zone or wetting the foliage.

End of production data were collected 1 day after the last application of Si treatments and consisted of measuring plant canopy height and width, and stem diameter. Canopy height and width measurements were collected using methods described in Experiment #1. Stem diameter was measured 1 cm below the apical meristem for the tallest shoot and also 1 cm above the substrate surface per replicate.

Replicates were then irrigated to container capacity with deionized water and moved to a post-production room simulating an indoor retail environment. Low-intensity light was supplied using incandescent and fluorescent lighting fixtures, a 9-h photoperiod

was maintained, and ambient air temperature ranged from 20 to 22 °C. Plastic barriers were placed over the substrate for each replicate and fit around plant stems to minimize evaporation from the substrate surface. Evaporation pans were placed in each block, and water loss from each pan was recorded twice daily. The number of days until visible wilt was measured for each treatment replicate following the same methods used in Experiment #1.

Following post-production plants were rehydrated with clear water, and shoot growth and shoot tissue Si concentrations were measured for each replicate following methods described for Experiment #1. The weight of Si accumulated in shoots was determined by multiplying the Si concentration in the dried shoot tissue by the total shoot dry weight per replicate. Percent increase in shoot tissue Si as a result of the Si foliar spray and drench treatments was determined for each species by dividing the shoot Si concentration and accumulated shoot Si values per replicate by the average for the non-Si controls, then multiplying by 100%. More than 90% of the Si taken up by plants roots is reportedly translocated to the shoots [2], and root Si accumulation was not evaluated in this study.

Analysis of variance (ANOVA) using PROC GLIMMIX (SAS version 9.4; The SAS Institute, Cary, NC, USA) was used to evaluate plant species and supplemental Si treatment effects on canopy height and width, stem diameter, shoot dry weight, shoot tissue Si, and percent increase in shoot Si. Means separation used Tukey's honestly significant difference (hsd) at $\alpha = 0.05$.

## 3. Results

### 3.1. Experiment #1: Evaluation of Si Foliar Spray Concentrations with Basil

Silicon foliar sprays ranging from 50 to 400 mg·L$^{-1}$ Si increased leaf SPAD chlorophyll content compared to the control treatment (0 mg·L$^{-1}$ Si) for container-grown basil (Tables 1 and 2). However, all plants had dark green foliage and appeared healthy, evidenced by high leaf SPAD values > 30 across treatments. Increased Si supply and uptake has been shown to increase the distribution and activity of micronutrients in leaf tissues for certain plant species [2], particularly iron and manganese, which are involved in the structure and functioning of chlorophyll. Although micronutrients were not analyzed in this experiment, it is possible Si foliar sprays increased leaf micronutrient activity and contributed to the darker green foliage color.

**Table 1.** Statistical parameters for silicon (Si) foliar sprays (50, 100, 200, 400 mg·L$^{-1}$ Si) compared to a 0 mg·L$^{-1}$ Si control applied every 7 days with container-grown basil. Leaf SPAD chlorophyll content, shoot dry weight, shoot tissue Si concentration, and number of days until visible wilt were measured.

| | Silicon Foliar Treatment (mg·L$^{-1}$ Si) | F-Statistic [z] | *p*-Value |
|---|---|---|---|
| Leaf SPAD chlorophyll content | 50 | 26.68 | 0.0001 |
| | 100 | 17.05 | 0.0009 |
| | 200 | 19.52 | 0.0005 |
| | 400 | 15.14 | 0.0014 |
| Shoot dry weight (g) | 50 | 1.66 | 0.2177 |
| | 100 | 0.68 | 0.4239 |
| | 200 | 2.36 | 0.1451 |
| | 400 | 4.60 | 0.0488 |
| Shoot Si concentration (mg·kg$^{-1}$) | 50 | 0.60 | 0.4514 |
| | 100 | 13.59 | 0.0022 |
| | 200 | 78.93 | <0.0001 |
| | 400 | 146.16 | <0.0001 |
| Number of days until wilting | 50 | 2.22 | 0.1569 |
| | 100 | 4.19 | 0.0586 |
| | 200 | 4.60 | 0.0488 |
| | 400 | 5.81 | 0.0292 |

[z] Single degree-of-freedom contrasts were based on 4 and 8 replicates for each Si foliar spray treatment (50, 100, 200, 400 mg·L$^{-1}$ Si) and the 0 mg·L$^{-1}$ Si control treatment, respectively.

**Table 2.** Silicon (Si) foliar spray treatment effects on leaf SPAD chlorophyll content, shoot dry weight, and shoot tissue Si concentration at the end of production for container-grown basil, and number of days until visible wilt in a simulated indoor retail environment.

| Si Foliar Treatment (mg·L$^{-1}$ Si) | Leaf SPAD Chlorophyll Content | | Shoot Dry Weight (g) | | Shoot Si Concentration (mg·kg$^{-1}$) | | Number of Days until Wilting | |
|---|---|---|---|---|---|---|---|---|
| 0 | 34.2 [z] | | 5.5 | | 466.3 | | 11.6 | |
| 50 | 41.8 | *** [y] | 6.4 | NS | 439.6 | NS | 13.1 | NS |
| 100 | 40.3 | ** | 6.1 | NS | 593.2 | * | 13.7 | NS |
| 200 | 40.7 | ** | 6.6 | NS | 772.3 | *** | 13.8 | * |
| 400 | 40.0 | ** | 7.0 | * | 882.7 | *** | 14.1 | * |

[z] Data represent least-square means of four replicates for the individual Si foliar spray treatments (50, 100, 200, 400 mg·L$^{-1}$ Si) and eight replicates for the 0 mg·L$^{-1}$ Si control. Single degree-of-freedom contrasts were used to compare each Si foliar treatment to the control. [y] NS, *, **, *** Non-significant or significant at $p \leq 0.05$, 0.01, or 0.0001, respectively, for comparison of each Si foliar spray treatment (50, 100, 200, 400 mg·L$^{-1}$ Si) to the 0 mg·L$^{-1}$ Si control. See Table 1 for *p*-values.

Silicon foliar sprays had no effect on basil canopy height, canopy width, or shoot fresh weight compared to the control (data not shown). The 400 mg·L$^{-1}$ Si spray treatment did increase shoot dry weight at harvest (Tables 1 and 2); however, this increase in shoot growth was barely significant ($p = 0.0488$, Table 1). In addition, the increase in dry weight did not correspond to a lower relative water content (data not shown). Previous studies have reported increased growth and yield with added Si for certain plant species [3,10]; however, these results indicate added Si has minimal effects on growth for container-grown basil, and the potential increase in dry weight needs further investigation and validation.

Silicon foliar sprays at 100 mg·L$^{-1}$ Si or greater increased shoot tissue Si concentrations for basil at the end of production (Tables 1 and 2). Foliar sprays of 50 mg·L$^{-1}$ Si did not increase shoot tissue Si concentrations whereas foliar sprays of 400 mg·L$^{-1}$ Si nearly doubled shoot tissue Si compared to the control. Similar observations were reported by Kamenidou et al. [12], who reported increased leaf Si for container-grown zinnia and sunflower with sodium silicate sprays applied every 7 days at 150 mg·L$^{-1}$ Si. Basil sprayed with the 0 mg·L$^{-1}$ Si control treatment also accumulated a significant amount of Si (466.3 mg·kg$^{-1}$ Si; Table 2), indicating trace amounts of Si were likely supplied by the spray solution surfactant, growing substrate, water-soluble fertilizer, and/or water source.

Silicon foliar sprays at $\geq$200 mg·L$^{-1}$ Si increased the number of days until visible wilt for basil during post-production in the simulated retail environment (Tables 1 and 2). Basil sprayed with 200 and 400 mg·L$^{-1}$ Si wilted after 13.8 and 14.1 d, respectively, resulting in a 2.2 and 2.5 days increase compared to the control (Table 2). Supplemental Si applications with container-grown floriculture species have also been shown to increase plant resistance to wilting in retail [9].

*3.2. Experiment #2: Edible Plant Species Supplied with Si Foliar Sprays and Substrate Drenches*

Silicon foliar and substrate drench treatments had the greatest impact on plant morphology and growth for parsley (Table 3, Figure 1), with minimal or no effects on plant performance for the remaining crop species. Canopy height and width were not impacted by Si treatment for any species (Table 3), although a nearly significant reduction in canopy height occurred for parsley compared to the control ($p = 0.0772$). Stem diameter at the shoot tip increased for parsley supplied with supplemental Si (Table 3), whereas stem diameter at the base of each plant was not influenced by Si treatment for any species. For basil, the Si drench resulted in a greater shoot dry weight compared to the Si foliar spray (Table 3), where shoot dry weight for the control was intermediate. Parsley had greater shoot dry weights for both Si foliar spray and drench treatments compared to the control, and shoot growth was not influenced by Si treatment for the other plant species (Table 3).

**Table 3.** Effects of supplemental silicon (Si) treatments on canopy height and width, stem diameter at the shoot tip and base of the plant, and shoot dry weight measured at the end of production for six container-grown edible species. Silicon foliar sprays and substrate drench treatments were applied every 7 days during production.

| Species | Supplemental Si Treatment | Canopy Height (cm) | | Canopy Width (cm) | | Stem Diameter at Shoot Tip (mm) | | Stem Diameter at Plant Base (mm) | | Shoot Dry Weight (g) | |
|---|---|---|---|---|---|---|---|---|---|---|---|
| Basil | Control (0 mg·L$^{-1}$ Si) | 29.3 | a | 33.7 | a | 3.1 | a | 5.6 | a | 4.10 | ab |
| | Si drench (100 mg·L$^{-1}$ Si) | 32.7 | a | 34.2 | a | 2.8 | a | 6.0 | a | 4.47 | a |
| | Si spray (500 mg·L$^{-1}$ Si) | 27.3 | a | 31.7 | a | 2.7 | a | 5.9 | a | 3.52 | b |
| | | NS | | NS | | NS | | NS | | * | |
| Cucumber | Control (0 mg·L$^{-1}$ Si) | 64.3 | a | 40.3 | a | 3.5 | a | 8.3 | a | 6.10 | a |
| | Si drench (100 mg·L$^{-1}$ Si) | 58.7 | a | 42.7 | a | 3.7 | a | 8.0 | a | 6.44 | a |
| | Si spray (500 mg·L$^{-1}$ Si) | 63.0 | a | 41.7 | a | 3.3 | a | 7.9 | a | 5.92 | a |
| | | NS | | NS | | NS | | NS | | NS | |
| Parsley | Control (0 mg·L$^{-1}$ Si) | 5.0 | a | 41.0 | a | 5.8 | b | 12.0 | a | 3.15 | b |
| | Si drench (100 mg·L$^{-1}$ Si) | 2.3 | a | 48.2 | a | 10.2 | a | 13.3 | a | 5.09 | a |
| | Si spray (500 mg·L$^{-1}$ Si) | 3.0 | a | 46.8 | a | 9.1 | ab | 13.0 | a | 4.60 | a |
| | | NS | | NS | | * | | NS | | * | |
| Rosemary | Control (0 mg·L$^{-1}$ Si) | 11.7 | a | 13.2 | a | 1.8 | a | 3.3 | a | 0.92 | a |
| | Si drench (100 mg·L$^{-1}$ Si) | 11.7 | a | 11.2 | a | 1.9 | a | 2.7 | a | 0.83 | a |
| | Si spray (500 mg·L$^{-1}$ Si) | 15.7 | a | 13.0 | a | 1.8 | a | 3.1 | a | 1.01 | a |
| | | NS | | NS | | NS | | NS | | NS | |
| Thyme | Control (0 mg·L$^{-1}$ Si) | 19.7 | a | 23.2 | a | 1.0 | a | 2.7 | a | 1.71 | a |
| | Si drench (100 mg·L$^{-1}$ Si) | 17.0 | a | 21.5 | a | 0.9 | a | 2.5 | a | 1.73 | a |
| | Si spray (500 mg·L$^{-1}$ Si) | 14.7 | a | 23.0 | a | 1.0 | a | 2.4 | a | 1.43 | a |
| | | NS | | NS | | NS | | NS | | NS | |
| Tomato | Control (0 mg·L$^{-1}$ Si) | 51.7 | a | 53.2 | a | 2.3 | a | 7.9 | | 7.68 | a |
| | Si drench (100 mg·L$^{-1}$ Si) | 50.0 | a | 54.0 | a | 2.5 | a | 7.4 | | 7.91 | a |
| | Si spray (500 mg·L$^{-1}$ Si) | 54.3 | a | 56.0 | a | 2.2 | a | 7.5 | | 7.48 | a |
| | | NS | | NS | | NS | | NS | | NS | |

NS, * Nonsignificant or significant at $p \le 0.05$.

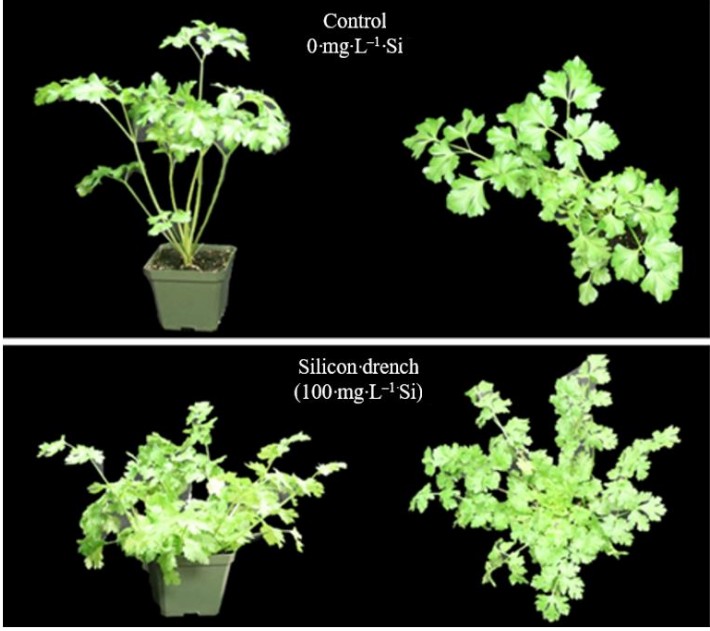

**Figure 1.** Effects of a no supplemental silicon (Si) control versus a Si substrate drench treatment on plant growth and morphology of parsley. Left and right photos consist of side and top views, respectively. The visual appearance of parsley was not different between the Si drench and Si foliar spray treatment (not shown).

Silicon foliar spray and drench treatments influenced the overall visual appearance for parsley, as shown in Figure 1, whereas there were no differences in visual appearance for the remaining species (data not shown). Parsley had visible reductions in leaf expansion, internode length, and petiole length, resulting in more of a rosette-like appearance (Figure 1). Leaf number was not measured but appeared greater for Si-treated plants versus the control (Figure 1), which may have partially explained the increased shoot dry weights in Table 3. Leaf margins became more sharply serrated with Si-treated plants compared to the more lobular margins observed in control plants. Necrosis also developed around the margins of young expanding leaves for Si-treated plants.

Plant species differed in shoot tissue Si concentrations for each Si treatment as shown in Table 4. Cucumber consistently had the greatest shoot tissue Si (Table 4), and supplemental Si treatments increased shoot Si levels for all species. Parsley had the lowest shoot Si concentrations for the control and Si drench treatments, whereas tomato was lowest for the Si foliar spray treatment. Similar to observations in Experiment #1, all plant species accumulated a measurable amount of Si with the 0 mg·L$^{-1}$ Si control treatment, indicating trace amounts of Si were taken up from the substrate, fertilizer, spray surfactant, and/or irrigation water sources. Shoot Si concentrations for basil supplied with control treatments were similar between Experiments #1 and #2, as shown in Tables 2 and 4 (466.3 mg·kg$^{-1}$ Si for Experiment #1 and 459.7 mg·kg$^{-1}$ Si for Experiment #2), indicating consistency in accumulation of trace amounts of Si for basil.

**Table 4.** Supplemental silicon (Si) treatment effects on Si concentration and total accumulated Si in dried shoot tissues at the end of production for six container-grown edible species. Silicon foliar spray and substrate drench treatments were applied every 7 days during production.

| Plant Species | Control (0 mg·L$^{-1}$ Si) | | Si Foliar Spray (500 mg·L$^{-1}$ Si) | | Si Substrate Drench (100 mg·L$^{-1}$ Si) | |
|---|---|---|---|---|---|---|
| | Si concentration in shoot tissue (mg·kg$^{-1}$) | | | | | |
| Basil | 459.7 [z] | b | 593.5 | b | 943.0 | b |
| Cucumber | 1095.3 | a | 1577.3 | a | 2838.4 | a |
| Parsley | 124.9 | c | 494.5 | b | 342.3 | c |
| Rosemary | 142.3 | bc | 359.9 | b | 351.6 | c |
| Thyme | 232.1 | bc | 641.9 | b | 656.0 | bc |
| Tomato | 254.4 | bc | 343.4 | b | 578.6 | bc |
| | Accumulated Si in shoot tissue (mg/plant) | | | | | |
| Basil | 1.85 | b | 2.09 | bc | 4.21 | bc |
| Cucumber | 6.66 | a | 0.32 | a | 18.28 | a |
| Parsley | 0.40 | cd | 2.28 | b | 1.75 | cd |
| Rosemary | 0.13 | d | 0.37 | d | 0.29 | d |
| Thyme | 0.40 | cd | 0.92 | cd | 1.16 | d |
| Tomato | 1.96 | b | 2.54 | b | 4.57 | b |

[z] Data represent least-square means of three replicates, analyzed by Si treatment with means separation using Tukey's honestly significant difference (hsd) at $\alpha = 0.05$.

Total accumulation of Si in shoots followed a similar trend to that observed for shoot Si concentration (Table 4), where cucumber had the greatest Si accumulation across species and treatments. Overall, rosemary and thyme had the lowest accumulation of Si compared to the other species (Table 4), in part because these species also had the least amount of shoot growth during the experiment (Table 3).

Shoot Si concentrations increased for most species as a result of the Si foliar spray and drench treatments (Figure 2A). Silicon drench treatments increased shoot Si concentrations by 105% for basil to 183% for parsley (Figure 2A), and therefore at least doubled shoot Si across species. Silicon foliar sprays increased shoot tissue Si by 29% for basil to 396% for parsley (Figure 2A). The 95% confidence intervals in Figure 2A overlapped the *x*-axis for basil, cucumber, and tomato supplied with the Si foliar spray, suggesting that increased shoot Si for these treatments was not statistically different from 0%. Basil, cucumber, and tomato also had the greatest shoot growth and canopy size (Table 3), and greater

overlapping of leaves may have reduced the Si foliar spray coverage and efficacy for these species. Percent increases in total Si accumulation in shoots followed similar trends to those observed in Figure 2A for increases in shoot Si concentration (Figure 2B).

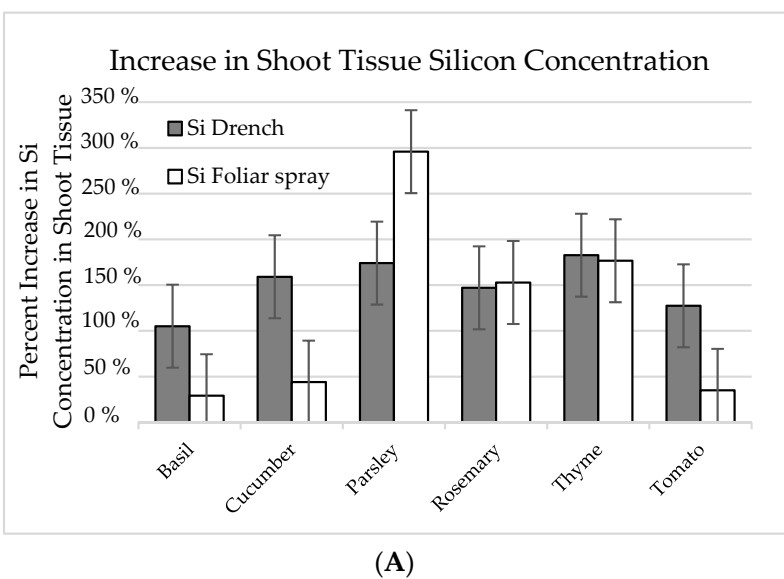

(**A**)

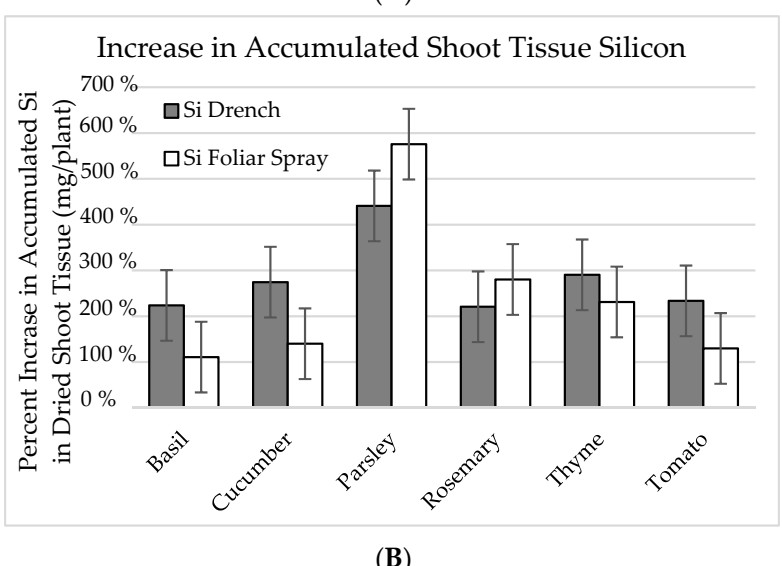

(**B**)

**Figure 2.** Silicon (Si) treatment effects on the percent increase in shoot tissue Si (**A**) and accumulated Si (**B**) relative to the control treatments at the end of production for six container-grown edible species. Silicon foliar spray (500 mg·L$^{-1}$ Si), substrate drench (100 mg·L$^{-1}$ Si), and control (0 mg·L$^{-1}$ Si) treatments were applied every 7 days during production. Data represent least-square means of three replicates, and error bars represent 95% confidence intervals.

The efficiency of Si uptake into shoot tissues was estimated by dividing the total Si accumulation in shoots by the total amount of Si supplied per replicate, and species data were analyzed by Si treatment using ANOVA (Table 5). Total supplied Si included Si measured in the water-soluble fertilizer, irrigation water source, applied foliar sprays and substrate drenches, but did not include Si released over time from the substrate. Total Si supplied was (in mg per replicate) 6.9 for the control and 60.9 for the Si drench, and ranged from 9.0 to 28.2 for the Si foliar sprays (see Materials and Methods).

**Table 5.** Supplemental silicon (Si) treatment effects on percent Si uptake efficiency in shoot tissues for six container-grown edible species. Silicon foliar spray and substrate drench treatments were applied every 7 days during production.

| Plant Species | Control (0 mg·L$^{-1}$ Si) | | Si Foliar Spray (500 mg·L$^{-1}$ Si) | | Si Substrate Drench (100 mg·L$^{-1}$ Si) | |
|---|---|---|---|---|---|---|
| | Si uptake efficiency in shoot tissue (%) | | | | | |
| Basil | 27.3 [z] | b | 8.0 | b | 6.3 | b |
| Cucumber | 95.6 | a | 27.2 | a | 27.3 | a |
| Parsley | 6.0 | c | 8.7 | b | 2.3 | c |
| Rosemary | 2.0 | c | 2.3 | c | 0.4 | c |
| Thyme | 5.7 | c | 5.0 | bc | 1.7 | c |
| Tomato | 28.3 | b | 7.3 | b | 6.7 | b |
| | *** [y] | | *** | | *** | |

[z] Data represent least-square means of three replicates, analyzed by Si treatment with means separation using Tukey's honestly significant difference (hsd) at α = 0.05. [y] *** Significant at $p \leq 0.0001$.

Cucumber consistently had the greatest Si uptake efficiency across treatments (Table 5) and took up nearly all Si supplied in the control (95.6% of total supplied Si). Silicon uptake efficiency was <10% of total Si supplied with the foliar spray and drench treatments, except for cucumber, where uptake efficiency was approximately 27% (Table 5). These results indicated the majority of Si supplied in the foliar spray and drench treatments was not taken up into shoots during this experiment. With foliar treatments, a potassium silicate residue was visible on basil, cucumber, and tomato leaves (data not shown), although the amount of Si deposited on leaf surfaces was not quantified. Overall, there were strong linear-positive correlations in species Si uptake efficiency between the control and the Si foliar spray and drench treatments [r > 0.95, (data not shown)], which suggested that species maintained their Si uptake efficiency regardless of the Si treatment and the total amount of Si supplied.

Silicon foliar and drench treatments had no effect on resistance to wilting in this experiment (data not shown), whereas in Experiment #1 basil supplied with Si foliar sprays at 200 and 400 mg·L$^{-1}$ Si showed an increase in number of days until wilt during post-production (Table 2). It is possible the sodium silicate used for foliar sprays in Experiment #1 was more effective at delaying wilt compared to the potassium silicate used in Experiment #2. Kamenidou et al. [9] also reported weekly sodium silicate foliar sprays at 100 mg·L$^{-1}$ Si increased leaf resistance to water loss and reduced transpiration with zinnia, whereas weekly substrate drenches with potassium silicate had no effect. These authors hypothesized that the increase in leaf resistance resulted from an anti-transpirant film formed across leaf surfaces by the Si foliar sprays. Although the potassium silicate drenches increased shoot Si levels, Kamenidou et al. [9] suggested the drenches did not form an anti-transpirant film or result in a systemic Si-mediated increase in stomatal resistance reported for certain agronomic crop species [21,22].

It may be reasonable to expect the potassium silicate used in Experiment #2 to form an anti-transpirant film similar to sodium silicate when applied as a foliar spray. In addition, potassium silicate and other Si-containing products applied to the root zone, such as the substrate incorporation of rice hulls and calcium silicate slag, is also common in commercial floriculture and has been reported to delay wilting for consumers [16]. Inconsistent effects of Si on resistance to wilting may be caused by differences in environmental conditions as suggested by Kamenidou et al. [9], and particularly differences in plant stress between experiments. Further research may be needed to explore the interactions between growing conditions and cultural practices to better evaluate the potential benefits of Si with container-grown edibles.

## 4. Discussion

Supplemental Si has been shown to sometimes influence plant morphology and result in plant growth abnormalities with container-grown floriculture species [7–9]. Mattson

and Leatherwood [7] evaluated 18 bedding plant species supplied with weekly potassium silicate drenches at 100 mg·$L^{-1}$ Si and found Si drenches influenced the growth and morphology of 14 species and caused either increased or decreased plant height, stem diameter, flower diameter, leaf thickness, or shoot fresh and/or dry weights. Potassium silicate drenches above 100 mg·$L^{-1}$ Si were also found to decrease plant growth and result in flower deformation with container-grown gerbera, sunflower, and zinnia [8,9].

Kamenidou et al. [8–10] found Si drenches altered the elemental composition in dry plant tissues; weekly potassium silicate drenches at 100 mg·$L^{-1}$ Si decreased shoot tissue magnesium (Mg) for sunflower, zinnia, and gerbera, and drenches at $\geq$200 mg·$L^{-1}$ Si increased shoot tissue potassium (K) for the same species. However, there was no indication or visual symptoms of nutritional disorders, and Mg and K remained within typical nutrient sufficiency ranges (1.5 to 3.5 g·$kg^{-1}$ dry weight for Mg, and 20 to 50 g·$kg^{-1}$ dry weight for K; Marschner [2]). Supplemental Si effects on shoot tissue nutrients were not measured in Experiments #1 or #2, but they would be important to evaluate in future studies with container-grown edibles, particularly longer-term crops where Si treatment would be expected to have more of an impact.

Determining the underlying cause of the shoot distortion and leaf tip necrosis observed for Si-treated parsley was beyond the scope of this study and may deserve further investigation. The authors have found no current reports of Si toxicity, although it is possible that phytotoxicity was caused by potential carriers and/or additives in the commercial-grade potassium silicate product. The symptoms in Figure 1 also partially resembled the general symptoms of boron (B) and/or calcium (Ca) deficiency observed for agronomic and greenhouse crops [2,23]. The transport of Si, B, and Ca throughout plant tissues is typically passive via the bulk flow of water, and all three elements are deposited primarily within plant cell walls and have similar roles in providing cell wall stability [2]. The capacity for Si uptake is also negatively related to plant requirements for B and Ca uptake [24,25], where species low in Si uptake capacity have relatively high demand for B and Ca. The low shoot Si concentrations for parsley in Table 4 indicate a relatively low Si uptake capacity, and one possibility may be the additional Si taken up in response to the Si treatments increased nutrient competition and therefore induced B and/or Ca deficiency.

Past research investigating the role of Si in plant biology has led to the classification of plant species into two main categories: Si "accumulators" and "non-accumulators" [1], where "accumulators" and "non-accumulators" tend to accumulate Si at concentrations greater or less than 1000 mg·$kg^{-1}$, respectively. Table 4 suggests cucumber was the only Si "accumulator" in this study and that species classification was not impacted by Si treatment. Voogt and Sonneveld [3] also reported cucumber as an "accumulator" species, with supplemental Si resulting in increased yield and resistance to powdery mildew.

Certain "non-accumulator" species have also been shown to benefit from supplemental Si [3,26–28], despite negligible Si uptake. Tomato, lettuce (*Lactuca sativa* L.), and strawberry (*Fragaria* sp. Duchesne) are examples of "non-accumulator" species where supplemental Si does not increase yield but reduces susceptibility to micronutrient toxicity as well as powdery mildew [3]. Although the majority of container-grown edible species were classified as Si "non-accumulators" in this study, it is possible these species would still benefit from additional Si supplied during production, particularly for the prevention of diseases and micronutrient disorders.

## 5. Conclusions

Supplemental Si applied as a foliar spray and substrate drench increased shoot tissue Si for a range of container-grown edible species. Supplemental Si had minimal effects on plant growth and morphology for most species, with the exception of parsley, which developed distorted growth when potassium silicate was applied as a foliar spray and substrate drench at 500 and 100 mg·$L^{-1}$ Si, respectively. Silicon foliar sprays increased plant resistance to wilting in a simulated indoor retail environment for basil during the

first experiment, but this effect was not replicated for basil or any other species during the second experiment and may therefore need further investigation.

Silicon drenches tended to be more effective at increasing shoot Si concentrations compared to foliar sprays, particularly for species with greater amounts of shoot growth and larger canopy sizes. Parsley was the only species where shoot Si was greater with the foliar spray compared to the drench; however, changes in the morphology of parsley may have increased the capture of Si spray solution by the foliage and growing tip. The majority of supplemental Si applied was not taken up into the shoot tissues. In addition, all species accumulated measurable amounts of Si with the control (no Si) treatments, indicating trace amounts of Si were taken up from the growing substrate, water-soluble fertilizer, spray surfactant, and irrigation water.

Cucumber was classified as a Si "accumulator" species with a high capacity for Si uptake, whereas the remaining species were found to be "non-accumulators" (basil, parsley, thyme, rosemary, and tomato) with a low Si uptake capacity. However, "non-accumulator" species may still benefit from supplemental Si applications, potentially decreased susceptibility to certain micronutrient disorders and foliar pathogens. The effects of supplemental Si on parsley in this study emphasize the importance for growers to conduct trials prior to applying Si to the entire crop, and the decision to supplement Si as a foliar spray or drench would likely depend mostly on practicality and cost.

**Author Contributions:** Conceptualization J.B.T. and R.W.D.; methodology J.B.T. and R.W.D.; formal analysis J.B.T., L.L.H. and R.W.D.; investigation J.B.T., L.L.H. and R.W.D.; writing-original draft preparation J.B.T. and R.W.D.; writing-review and editing J.B.T., L.L.H. and R.W.D.; supervision R.W.D. All authors have read and agreed to the published version of the manuscript.

**Funding:** This research was funded by the University of Arkansas Dale Bumpers College of Agriculture and Life Sciences, University of Arkansas Honors College.

**Institutional Review Board Statement:** Not applicable.

**Informed Consent Statement:** Not applicable.

**Data Availability Statement:** The data presented in this study are available on request from the corresponding author, and are not publicly available as a way to maintain data integrity and prevent misuse.

**Acknowledgments:** We thank the Arkansas Division of Agriculture, Arkansas Agricultural Experiment Station, the U.S. Department of Agriculture (USDA) National Institute of Food and Agriculture projects #1022864 and #1019001 for additional support. We also thank Kalyn Helms for help with data collection.

**Conflicts of Interest:** The authors declare no conflict of interest.

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
