# Peer review of "Silicon Foliar Spray and Substrate Drench Effects on Plant Growth, Morphology, and Resistance to Wilting with Container-Grown Edible Species"

_horticulturae, doi:10.3390/horticulturae7090263_

Round 1

Reviewer 1 Report

Figure 1. Effects of a no supplemental silicon (Si) control versus a Si substrate drench treatment on plant growth and morphology of parsely. Left and right photos consist of side and top views, respectively. The visual appearance of parsley was not different btween the Si drench and Si foliar spray treatment (not shown).

Figure 1.  Please correct the spelling of the word “between”.

Overall, this was an interesting and easy-to-read manuscript.  

Author Response

Response to review comments for submitted Horticulturae manuscript titled “Silicon foliar spray and substrate drench effects on plant growth, morphology, and resistance to wilting with container-grown edible species”

Manuscript ID: horticulturae-1344091

Reviewer #1

Figure 1. Effects of a no supplemental silicon (Si) control versus a Si substrate drench treatment on plant growth and morphology of parsely. Left and right photos consist of side and top views, respectively. The visual appearance of parsley was not different btween the Si drench and Si foliar spray treatment (not shown).

Figure 1. Please correct the spelling of the word “between”.

Overall, this was an interesting and easy-to-read manuscript.  

In the Figure 1 caption, spelling for the words “between” and “parsley” were corrected. We have also reread the manuscript to look for and correct other spelling errors.

Reviewer 2 Report

The work by Tebow et al describes a study into the effects of different silicon fertilisation regimes on growth and morphological aspects of a number of edible plant species. The work consists of two main experiments, the first using basil only while the second uses basil and 5 further species. The work presents some interesting and useful results such as the significant postponement of crop wilting after foliar spraying at 200 mg/L Si or higher. The paper is well written, the experimental work is competently executed and the data are appropriately analysed, however there is a problem with the methodology.

Introduction:

-please mention that (beneficial) Si effects are primarily observed in stress conditions and discuss the relevance of this regarding the work presented here

Methods:

-‘fertigated as needed’ ; this is very vague and should be further specified. For example, how did this vary between treatments (if at all)? Did this lead to different watering regimes?

-foliar treatments of 400/500 mg Si/L are equivalent to ~15/20 mM Si and ~15-25 mM Na or K, very high concentrations that will rise considerably further after some evaporation. Effectively, these plants are exposed to salt stress and (in the case of K) potentially also to extra nutrient supply. The authors should have used Na/KCl controls (rather than water) to compensate for these effects and I strongly suggest that at least changes in tissue Na and K are measured (since the authors have ICP samples they may still have enough to do this analysis since only a tiny volume is needed that can be diluted).

-as stated above, the used Si concentrations in the foliar spray are very high;  (A) does this lead to deposition on the leaves? (B) it would be good to have some idea of the fraction of Si that is actually taken up and that which is simply precipitated at the leave surface, possibly forming a physical barrier against evapotranspiration. (A wash with some CaCl2 and subsequent Si analysis of the extract could give some idea of this).

Results:

-it would be interesting to see an analysis of how the proportion of Si uptake (as % of supply) differs between roots and shoots (and by extension between foliar and soil application).

-exp 1 shows no significant effect of Si on plant FW but there is a (marginally significant) effect on DW. This implies the plants have lower relative water content. Was there a trend (the authors should have all the data from their wilting assays)?

Minor issues:

-‘ 1800 and 2000 HR’  presumably ‘18:00 and 20:00 h’ is meant here?

Author Response

substrate drench effects on plant growth, morphology, and resistance to wilting with container-grown edible species”

Manuscript ID: horticulturae-1344091

Reviewer #2

Comments and Suggestions for Authors

The work by Tebow et al describes a study into the effects of different silicon fertilisation regimes on growth and morphological aspects of a number of edible plant species. The work consists of two main experiments, the first using basil only while the second uses basil and 5 further species. The work presents some interesting and useful results such as the significant postponement of crop wilting after foliar spraying at 200 mg/L Si or higher. The paper is well written, the experimental work is competently executed and the data are appropriately analysed, however there is a problem with the methodology.

Introduction:

-please mention that (beneficial) Si effects are primarily observed in stress conditions and discuss the relevance of this regarding the work presented here

The first sentence of the Introduction was changed to the following: “Silicon (Si) is a major constituent of many mineral field soils, and although not considered an essential element in plant nutrition, has been reported to have beneficial effects on plant growth during crop production [1-3], particularly under conditions leading to plant stress.”

The following sentence was also added to the results and discussion section: “Inconsistent effects of Si on resistance to wilting may be caused by differences in environmental conditions as suggested by Kamenidou et al. [9], and particularly differences in plant stress between experiments.”

Methods:

-‘fertigated as needed’ ; this is very vague and should be further specified. For example, how did this vary between treatments (if at all)? Did this lead to different watering regimes?

This sentence in the Methods was changed to the following: “For the duration of the experiment, all treatment replicate plants were irrigated uniformly with fertilizer solution once substrate moisture content dropped to approximately 50% of container capacity, with a commercial water-soluble fertilizer (17-3-17 Peters Peat-Lite Special®; Everiss, Geldermalsen, The Netherlands) mixed at 150 mg∙L–1 nitrogen (N) in tap water with an electrical conductivity of <0.3 mS∙cm–1 and <60 mg∙L–1 bicarbonate alkalinity. Each replicate plant for each treatment received 1.35 L of fertilizer solution and 202.5 mg of N.”

-foliar treatments of 400/500 mg Si/L are equivalent to ~15/20 mM Si and ~15-25 mM Na or K, very high concentrations that will rise considerably further after some evaporation. Effectively, these plants are exposed to salt stress and (in the case of K) potentially also to extra nutrient supply. The authors should have used Na/KCl controls (rather than water) to compensate for these effects and I strongly suggest that at least changes in tissue Na and K are measured (since the authors have ICP samples they may still have enough to do this analysis since only a tiny volume is needed that can be diluted).

These comments are really helpful, and having Na/KCl controls would have helped isolate any direct effects of the Si as well as better understand the impacts of Si treatment on plant nutritional status. Unfortunately, we did measure tissue elements other than Si in this study, and are not able to retest the tissue collected. However, in the Discussion section we did include a new paragraph that discusses potential effects of potassium silicate on plant nutritional using examples from previous authors. The following was added:

“Kamenidou et al. [8-10] found Si drenches altered the elemental composition in dry plant tissues;  weekly potassium silicate drenches at 100 mg∙L–1 Si decreased shoot tissue magnesium (Mg) for sunflower, zinnia, and gerbera and drenches at ≥200 mg∙L–1 Si increased shoot tissue potassium (K) for the same species. However, there was no indication or visual symptoms of nutritional disorders, and Mg and K remained within typical nutrient sufficiency ranges (1.5 to 3.5 g∙kg–1 dry weight for Mg, and 20 to 50 g∙kg–1 dry weight for K; Marschner [2]). Supplemental Si effects on shoot tissue nutrients were not measured in Experiments #1 or #2, but would be important to evaluate in future studies with container-grown edibles, particuarly longer-term crops where Si treatment would be expected to have more of an impact.”

-as stated above, the used Si concentrations in the foliar spray are very high; (A) does this lead to deposition on the leaves? (B) it would be good to have some idea of the fraction of Si that is actually taken up and that which is simply precipitated at the leave surface, possibly forming a physical barrier against evapotranspiration. (A wash with some CaCl2 and subsequent Si analysis of the extract could give some idea of this).

The authors agree this would be really interesting, and in the future will likely look at fractions of Si taken up versus deposited on leaves. We did observe a film develop on basil, tomato, and cucumber leaves after foliar treatments, although there was no guarantee this was acting as a physical barrier. In the paragraph discussing Si uptake efficiency in the Results section, we changed the last couple sentences to acknowledge the development of a physical film, which now reads as follows:

“With foliar treatments, a potassium silicate residue was visible on basil, cucumber, and tomato leaves (data not shown), although the amount of Si deposited on leaf surfaces was not quantified. Overall, there were strong linear-positive correlations in species Si uptake efficiency between the control and the Si foliar spray and drench treatments [r > 0.95, (data not shown)], which suggested that species maintained their Si uptake efficiency regardless of the Si treatment and the total amount of Si supplied.”

Results:

-it would be interesting to see an analysis of how the proportion of Si uptake (as % of supply) differs between roots and shoots (and by extension between foliar and soil application).

This would be very interesting to measure in future experiments. Unfortunately, we did not measure root mass or Si content in this study. This is the reason the authors discussed Si uptake as “shoot Si” throughout the article, to specify that Si was only measured in above-ground tissues. We have also read that the majority of Si taken up by roots is translocated to the shoots, and have added the sentence below to the Materials and Methods section for Experiment #2.

“More than 90% of the Si taken up by plants roots is reportedly translocated to the shoots [2], and root Si accumulation was not evaluated in this study.”

-exp 1 shows no significant effect of Si on plant FW but there is a (marginally significant) effect on DW. This implies the plants have lower relative water content. Was there a trend (the authors should have all the data from their wilting assays)?

The statistical increase in dry weight did not result in a lower relative water content. Because fresh weight and water content were not affected by Si treatment, we feel that the marginal increase in dry weight at the highest Si spray rate needs further investigation and validation. Therefore, this additional information was added to the paragraph discussing fresh and dry weight gains, which now reads as follows:

“Silicon foliar sprays had no effect on basil canopy height, canopy width, or shoot fresh weight compared to the control (data not shown). The 400 mg∙L–1 Si spray treatment did increase shoot dry weight at harvest (Tables 1 and 2); however, this increase in shoot growth was barely significant (P=0.0488, Table 1). In addition, the increase in dry weight did not correspond to a lower relative water content (data not shown). Previous studies have reported increased growth and yield with added Si for certain plant species [3,10]; however, these results indicate added Si has minimal effects on growth for container-grown basil, and the potential increase in dry weight needs further investigation and validation.”

Minor issues:

-‘ 1800 and 2000 HR’  presumably ‘18:00 and 20:00 h’ is meant here?

The times 1800 and 2000 HR were changed to “18:00 and 20:00 h” and similar edits were made elsewhere in the text.

Round 2

Reviewer 2 Report

n/a